# Wheat Resistance to Stripe and Leaf Rusts Conferred by Introgression of Slow Rusting Resistance Genes

**DOI:** 10.3390/jof7080622

**Published:** 2021-07-31

**Authors:** Reda Ibrahim Omara, Atef Abdelfattah Shahin, Shaimaa Mahmoud Ahmed, Yasser Sabry Mostafa, Saad Abdulrahman Alamri, Mohamed Hashem, Mohsen Mohamed Elsharkawy

**Affiliations:** 1Wheat Diseases Research Department, Plant Pathology Research Institute, Agricultural Research Center, Giza P.O. Box 12619, Egypt; redaomara43@gmail.com (R.I.O.); a.a.shahin@hotmail.com (A.A.S.); 2ICARDA Biotechnology Lab, Agricultural Genetic Engineering Research Institute (AGERI), Giza P.O. Box 12619, Egypt; shaimaa2015@hotmail.com; 3Department of Biology, College of Science, King Khalid University, Abha P.O. Box 9004, Saudi Arabia; ysolhasa1969@hotmail.com (Y.S.M.); amri555@yahoo.com (S.A.A.); drmhashem69@yahoo.com (M.H.); 4Prince Sultan Bin Abdulaziz Center for Environmental and Tourism Research and Studies, King Khalid University, Abha P.O. Box 9004, Saudi Arabia; 5Department of Botany and Microbiology, Faculty of Science, Assiut University, Assiut P.O. Box 71515, Egypt; 6Agricultural Botany Department, Faculty of Agriculture, Kafrelsheikh University, Kafr Elsheikh P.O. Box 33516, Egypt

**Keywords:** wheat, rusts, resistant cultivars, molecular markers, breeding for resistance

## Abstract

Twenty-three wheat genotypes were evaluated for stripe and leaf rusts, caused by *Puccinia striiformis* f. sp. *tritici* and *Puccinia triticina* f. sp. *tritici*, respectively, at seedling and adult stages under greenhouses and field conditions during the 2019/2020 and 2020/2021 growing seasons. The race analysis revealed that 250E254 and TTTST races for stripe and leaf rusts, respectively were the most aggressive. Eight wheat genotypes (Misr-3, Misr-4, Giza-171, Gemmeiza-12, *Lr34/Yr18*, *Lr37/Yr17*, *Lr46/Yr29,* and *Lr67/Yr46*) were resistant to stripe and leaf rusts at seedling and adult stages. This result was confirmed by identifying the resistance genes: *Lr34/Yr18*, *Lr37/Yr17*, *Lr46/Yr29,* and *Lr67/Yr46* in these genotypes showing their role in the resistance. Sids-14 and Shandweel-1 genotypes were susceptible to stripe and leaf rusts. Twelve crosses between the two new susceptible wheat genotypes and the three slow rusting genes (*Lr34/Yr18*, *Lr37/Yr17,* and *Lr67/Yr46*) were conducted. The frequency distribution of disease severity (%) in F_2_ plants of the twelve crosses was ranged from 0 to 80%. Resistant F_2_ plants were selected and the resistance genes were detected. This study is important for introducing new active resistance genes into the breeding programs and preserving diversity among recently released wheat genotypes.

## 1. Introduction

Wheat (*Triticum aestivum* L.) is widely cultivated all over the world as a staple food. Wheat stripe and leaf rusts are caused by *Puccinia striiformis* f. sp. *tritici* and *Puccinia triticina* f. sp. *tritici*, respectively. They are the most common wheat diseases due to high yield losses and poor grain quality in susceptible wheat cultivars, especially at late growing dates [1,2].

Breeding programs are one of the most cost-effective strategies to control stripe and leaf rusts. Wheat breeding programs all over the world are incorporating rust resistance genes into commercial cultivars [3]. Genetic diversity in crops, especially wheat, is essential in the breeding program to improve the resistance of genotypes to biotic and abiotic stress conditions. A successful wheat breeding program depends mainly on the types and genetic variability available in wheat genotypes. The genetic variability is the most important natural resource in providing the required traits to develop new cultivars [4].

Breeders can use genetic diversity within adapted lines to choose parents for hybrid development with the most heterosis and incorporate the desired genes in a suitable background. Compared to broad genetic diversity, a limited genetic base is a serious impediment to breeding for various biotic and abiotic stresses. The limited genetic base of plant germplasm is a matter of concern for many modern plant breeding programs. However, wheat cultivars developed with larger genetic bases effectively improve the yield under several agro-climatic environments and resist the dissemination of diseases in new released cultivars [5].

To date, over 74 leaf rust resistance genes (Lr,s) have been identified, most of them, are mapped on different chromosomes [6]. The sudden appearances of new virulent races of the target pathogen have reduced the effectiveness of a significant number of rust resistance genes. Thus, stacking different resistant genes for stripe and leaf rusts in a given cultivar, a process called gene pyramiding, helps avoid the rapid breakdown of resistance and consequently achieve the durability of such resistance [7].

In Egypt, some newly produced wheat cultivars were quickly discarded after widespread cultivation due to their susceptibility to rusts under field conditions. While, other cultivars have been used in agriculture for many years, demonstrating suitable and high rust resistance. During a disease outbreak, most of these cultivars were known for their ability to impede rusts epidemics and, therefore, reducing disease epidemic rates.

Several kinds of molecular markers analyses have also been developed for genetic analysis of wheat populations [8]. The genomic structure composition that identifies essential genes for particular traits and preserves genetic materials for use in plant breeding was improved by genetic diversity studies using molecular markers [9]. Molecular markers, based on simple sequence repeats (SSR), are most frequently used for analyzing genetic diversity, particularly in cereals. These markers appear to be more informative in wheat than any other marker techniques since they show high polymorphism, co-dominant inheritance, and good reproducibility [10].

Using resistant cultivars to manage these serious diseases is the most efficient and environmentally friendly approach. Incorporating resistance genes into adapted germplasms, is a major goal in most wheat resistance breeding programs. Therefore, the major objectives were to study the response of 23 wheat genotypes to stripe and leaf rusts at seedling and adult stages under greenhouses and field conditions, as well as to produce and identify three combinations of resistant genes to stripe and leaf rusts on certain wheat genotypes.

## 2. Materials and Methods

Evaluation of 23 wheat genotypes was conducted: (i) At seedling stage in the greenhouse, Wheat Dis. Res. Dep., Plant Pathol. Res. Institute, Agricultural Research Center (ARC), (ii) at an adult stage in the experimental farm of ARC, Sakha, Kafr El-Sheikh governorate, during the 2019/20 and 2020/21 growing seasons. The molecular analysis was carried out in ICARDA Biotechnology Lab, AGERI, Egypt.

### 2.1. Evaluation of Wheat Genotypes Response to Stripe and Leaf Rusts

#### 2.1.1. At Seedling Stage

The response of 23 wheat genotypes was evaluated against stripe and leaf rusts at the seedling stage under greenhouse conditions (Table 1). They were evaluated against the most virulent and frequent races of the stripe and leaf rust pathogens; (6E4, 159E255, and 250E254) and (STSJT, MTTGT, and TTTST), respectively. Ten seeds from each tested wheat genotype were grown in plastic pots (6 cm in diameter). Each pot contains a mixture of soil and peat at a ratio of 1:1 (*v*:*v*). Seven-day-old seedlings of the 23 tested wheat genotypes were inoculated by brushing with urediniospores. Wheat genotypes were evaluated for stripe rust in the greenhouse of Agric. Res. Station, Sakha. While, they were evaluated for leaf rust in the greenhouse of Wheat Diseases Res. Dept., Plant Pathol. Res. Inst., ARC, Giza. The procedures for inoculation were performed following the methods described by Stakman et al. [11]. Twelve days after planting, a rust reaction was reported. The data of stripe rust were scored as an infection type, i.e., 0, 1, 2, 3, 4, 5, and 6 were considered resistant, while 7, 8, and 9 were susceptible [12]. The data of leaf rust were scored as an infection type, i.e., 0, 0, 1, and 2 were considered resistant, while 3 and 4 were susceptible [11].

#### 2.1.2. At Adult Stage

The same genotypes were evaluated under field conditions during the 2019/20 and 2020/21 growing seasons. A complete randomized block design with three replications was used. The experimental unit involved three rows (3 m long and 30 cm apart and 5 g seed rate for each row). The experiment was surrounded by 1 m allay and 1.5 m belts, served as a spreader of stripe and leaf rusts susceptible entries, i.e., “Morocco and *Triticum spleta saharences*”. Artificial inoculation of the spreader was done using a mixture of the three physiological races of stripe and leaf rust pathogens (6E4, 159E255, and 250E254) and (STSJT, MTTGT, and TTTST), respectively during tillering and elongation stages.

During two consecutive seasons, disease severity was rescored four times (at 10-day intervals) and represented as a percent leaf area covered with rust pustules. Immune (0), resistant (R), moderately resistant (MR), moderately susceptible (MS), and susceptible (S) were the five forms of rust reaction [11]. The obtained data served in the determination of the final rust severity (FRS%), as outlined by Das et al. [13]. The area under disease progress curve (AUDPC) was calculated for each genotype according to an equation proposed by Shaner and Finney [14] as follows:AUDPC = D [1/2 (Y_1_ + Y_K_) + Y_2_ + Y_3_ + … Y _(K−1)_]
where D = time intervals (days between consecutive records), Y_1_ + Y_k_ = sum of the first and the last disease scores, and Y_2_ + Y_3_ + …. + Y _(K−1)_ = sum of all the in between disease scores.

### 2.2. Molecular Markers Experiment

#### 2.2.1. DNA Isolation and PCR Protocol

Isolation of DNA and PCR protocol were carried out in ICARDA Biotechnology Lab, AGERI, Egypt. The procedures described by Rogers and Bendich [15] were used to extract DNA from green leaves of seedlings (5–7 days old). The PCR reaction mixture (10 μL) consisted of DNA template (5 ng), forward and reverse primers (10 pmol), and COSMO PCR Master Mix (Willowfort). PCR conditions were initiated by denaturation for 5 min at 94 °C, followed by 35 cycles (denaturation for 30 s at 94 °C, annealing for the 30 s at 55 °C for *Lr34/Yr18*, 51 °C for *Lr37/Yr17,* and 60 °C for *Lr46/Yr29* and *Lr67/Yr46*, and extension at 72 °C for 30 s) and final extension at 72 °C for 7 min. PCR products of SSR markers were loaded onto 2.5% agarose gel. Table 2 shows the primer sequences used to define rust resistance genes.

#### 2.2.2. Introgression of Resistant Genes for Stripe and Leaf Rusts in Wheat Genotypes

The resistance genes (*Lr34/Yr18*, *Lr37/Yr17,* and *Lr67/Yr46*) were introgressed into the wheat genotypes (Sids-14 and Shandweel-1) as follows: (Sids-14 × *Lr34/Yr18*, Sids-14 × *Lr37/Yr17,* and Sids-14 × *Lr67/Yr46*) and (Shandweel-1 × *Lr34/Yr18*, Shandweel-1 × *Lr37/Yr17,* and Shandweel-1 × *Lr67/Yr46*). All of the genotypes were grown in three different sowing dates. Crosses were carried out using two wheat genotypes as mother plants. The F_1_ seeds were harvested and kept for growing F_1_ plants in the next seasons (2020/21) in rows of 4 m long and 30 cm apart, and spaced 30 cm in order to allow the production of F_2_ seeds.

Seeds of F_1_ plants were sown individually in the subsequent growing seasons (2020/2021), so that each progeny could be examined to estimate their distribution frequencies. All of the plots were surrounded by a spreader area using a mixture of two highly susceptible wheat varieties, i.e., *Triticum spelta saharensis* and Morocco. The spreader wheat plants were moistened and dusted with uredospores-powder mixtures of the most prevalent races of stripe and leaf rust pathogens (6E4, 159E255, and 250E254) and (STSJT, MTTGT, and TTTST), respectively for inoculation in the field. Inoculation was performed at tillering and elongation stages [16]. Stripe and leaf rust severities (%) were recorded for each wheat plant of F_2_ generation at the first appearance of pustule. Under field conditions, F_2_ plants were divided into eight groups based on the severities of stripe and leaf rusts. Disease severity classes ranged from 0 to 10, 11 to 20, 21 to 30, 31 to 40, 41 to 50, 51 to 60, 61 to 70, and 71 to 80%. The first three classes were classified as having a low disease severity (resistant), while other classes (more than 30%) were considered as having a high disease severity (susceptible).

#### 2.2.3. Molecular Markers of F_2_ Plants

The resistant F_2_ plants were chosen for DNA isolation and PCR protocol in order to ensure that genes, *Lr34/Yr18*, *Lr37/Yr17,* and *Lr67/Yr46*, were transferred to F_2_ plants. DNA isolation and PCR protocol were done as mentioned in the previous method. The same primers of the three genes (Table 1) were used to ensure introgression of these genes into two wheat genotypes, Sids-14 and Shandweel-1.

### 2.3. Statistical Analysis

The analysis of variance (ANOVA) of the obtained data was performed with the software package SPSS18. The least significant difference (LSD) at a 5% level of significance was used to compare the treatment means.

## 3. Results

### 3.1. Evaluation of Genotypes against Stripe and Leaf Rusts at Seedling Stage

Responses of 23 wheat genotypes were evaluated against the most aggressive and frequent races of stripe rust pathogen (6E4, 159E255, and 250E254) and leaf rust pathogen (STSJT, MTTGT, and TTTST) (Figure 1A,B). As for the stripe rust, out of the 23 tested wheat genotypes, only 17 genotypes (Misr-3, Misr-4, Giza-139, Giza-168, Giza-171, Sakha-61, Sakha-94, Sakha-95, Gemmeiza-10, Gemmeiza-12, Sids-13, Sids-14, Shandweel-1, *Lr34/Yr18, Lr37/Yr17, Lr46/Yr29, and Lr67/Yr46*) were resistant to all the tested races and showed low infection type (Figure 1A). While, two wheat genotypes (Misr-1 and Misr-2) were susceptible against all the tested races and showed high infection type. This result may be attributed to the appearance of new virulent races such as 159E255 and 250E254, which are capable of supplanting the resistance of these genotypes. While, the responses of other wheat genotypes were different. The most aggressive race in supplanting the resistance in wheat genotypes (Misr-1, Misr-2, Giza-139, Giza-168, Sakha-94, Sakha-95, Gemmeiza-5, Gemmeiza-7, Gemmeiza-10, Sids-12, and Sids-13) was 250E254.

As for leaf rust, 23 wheat genotypes were tested against the most aggressive and frequent races of leaf rust pathogen, STSJT, MTTGT, and TTTST (Figure 1B). Out of the 23 tested, only nine wheat genotypes (Misr-3, Misr-4, Giza-171, Sakha-95, Gemmeiza-12, Sids-14, *Lr34/Yr18, Lr37/Yr17, and Lr67/Yr46*) were resistant to all the tested races and showed low infection type. While, three wheat genotypes (Giza-139, Sakha-61, and Gemmeiza-7) were susceptible against all the tested races and showed high infection type. On the other hand, the rest of the wheat genotypes showed different responses to the infection with the tested races. TTTST was the most aggressive race. It could supplant the resistance in 14 wheat genotypes.

### 3.2. Evaluation of Genotypes against Stripe and Leaf Rusts under Field Conditions

The final rust severity (FRS%) and AUDPC were studied at the adult stage during the 2019/20 and 2020/21 growing seasons. Results showed that the wheat genotypes (Misr-3, Misr-4, Giza-139, Giza-171, Sakha-61, Sakha-94, Gemmeiza-10, Gemmeiza-12, *Lr34/Yr18, Lr37/Yr17, Lr46/Yr29, and Lr67/Yr46*) exhibited the lowest final stripe rust severity and AUDPC values (Figure 2A,B). Misr-1, Misr-2, Gemmeiza-7, and Sids-12 showed the highest FRS (%) and AUDPC values, during the two seasons. The responses of other wheat genotypes (Giza-168, Sakha-95, Gemmeiza-7, Gemmeiza-9, Sids-13, Sids-14, and Shandweel-1) ranged from 17.33–40 for FRS% and 242.65 to 725 for AUDPC.

As for leaf rust, 16 wheat genotypes (Misr-1, Misr-2, Misr-3, Misr-4, Giza-168, Giza-171, Sakha-94, Sakha-95, Gemmeiza-12, Sids-12, Sids-13, Sids-14, *Lr34/Yr18, Lr37/Yr17, Lr46/Yr29, and Lr67/Yr46*) showed the lowest final leaf rust severity and AUDPC values during the two seasons (Figure 3A,B). Giza-139, Sakha-61, Gemmeiza-5, Gemmeiza-7, Gemmeiza-9, Gemmeiza-10, and Shandweel-1 exhibited the highest final rust severity and AUDPC values (Figure 3A,B).

It can be concluded that eight wheat genotypes (Misr-3, Misr-4, Giza-171, Gemmeiza-12, *Lr34/Yr18, Lr37/Yr17, Lr46/Yr29*, and *Lr67/Yr46*) were resistant to stripe and leaf rusts at seedling and adult stages, during the 2019/20 and 2020/21 growing seasons. Sakha-94 was resistant at the adult stage only during the two seasons.

### 3.3. Identification of Resistance Genes of Stripe and Leaf Rusts in Wheat Genotypes Using Molecular Markers

Resistance genes play an important role in wheat resistance to stripe and leaf rusts. Results of the present investigation clearly show the advantage of molecular markers for evaluating resistance genes in wheat genotypes compared to pedigree data.

Nineteen wheat genotypes (Misr-1, Misr-2, Misr-3, Misr-4, Giza-139, Giza-168, Giza-171, Sakha-61, Sakha-94, Sakha-95, Gemmeiza-5, Gemmeiza-7, Gemmeiza-9, Gemmeiza-10, Gemmeiza-12, Sids-12, Sids-13, Sids-14, and Shandweel-1) were used for detection of *Lr34/Yr18*, *Lr37/Yr18*, *Lr46/Yr29,* and *Lr67/Yr46* genes using molecular markers. The polymorphic survey revealed that out of the 19 wheat genotypes, the marker for *Lr34/Yr18* was identified as a fragment of 220 bp in 10 genotypes (Misr-3, Misr-4, Misr-2, Misr-1, Sids-13, Gemmeiza-12, Sakha-94, Giza-168, Gia-171, and Giza-139), while *Lr34/Yr18* was not found in nine genotypes (Sids-12, Sids-14, Sakha-61, Sakha-95, Gemmeiza-5, Gemmeiza-7, Gemmeiza-9, Gemmeiza-10, and Shandweel-1) (Figure 4). The *Lr34/Yr18* gene was characterized as a slow rusting resistance gene.

The marker of *Lr37/Yr17* was identified as a fragment of 285 bp in four genotypes, Misr-4, Misr-3, Gemmeiza-12, and Giza-171. This result explains the reason for resistance in these genotypes at seedling and adult stages for stripe and leaf rusts. *Lr37/Yr17* was not found in 15 genotypes, i.e., Misr-1, Misr-2, Giza-139, Giza-168, Sakha-61, Sakha-94, Sakha-95, Gemmeiza-5, Gemmeiza-7, Gemmeiza-9, Gemmeiza-10, Sids-12, Sids-13, Sids-14, and Shandweel-1 (Figure 5). Likewise, *Lr46/Yr29* was detected in all the tested genotypes except Gemmeiza-7 (Figure 6). Therefore, the 18 tested genotypes are carrying the resistance gene *Lr46/Yr29*. This gene was considered a slow rusting, which explains why certain genotypes are vulnerable to infection while still producing a high yield.

The marker for *Lr67/Yr46* was detected as a fragment of 198 bp in 16 genotypes (Misr-3, Misr-4, Misr-2, Misr-1, Gemmeiza-9, Gemmeiza-12, Sakha-94, Sakha-95, Giza-168, Gia-171, Giza-139, Sids-12, Sids-13, Sakha-61, Gemmeiza-5, and Gemmeiza-7). *Lr67/Yr46* was not found in three genotypes, Sids-14, Gemmeiza-10, and Shandweel-1 (Figure 7). It can be concluded that there are four genotypes (Misr-3, Misr-4, Giza-171, and Gemmeiza-12) carrying the four genes under study. They fell into one group through the polygenic tree (Figure 8). While, the genotypes, Sids-14, Sahndweel-1, and Gemmeiza-7 carry one gene and fall into another group through the polygenic tree (Figure 8).

These results illustrate the importance of pyramiding genes as a strategy to get longer-lasting tolerance with low genetic diversity, high gene flow, and asexual mating systems.

### 3.4. Introgression of Three Slow-Rusting Genes into the Two Wheat Genotypes

The resistance of newly released wheat genotypes such as Sids-14 and Shandweel-1 has rapidly lost its potency and these genotypes become susceptible in a short period due to the sudden emergency of the new and more aggressive races of the causal pathogens. *Lr46/Yr29*, a slow-rusting gene found in these genotypes, may be used in combination with other slow-rusting genes to produce elevated levels of APR to stripe and leaf rusts in wheat. Therefore, the wheat genotypes Sids-14 and Shandweel-1 with varying degrees of stripe and leaf rust severities and six F_2_ plants for each disease were used in this research. F_2_ plants were obtained from half diallel crosses between these genotypes and slow rusting genes *Lr34/Yr18*, *Lr37/Yr17*, and *Lr67/Yr46*. The obtained data are subjected to qualitative genetic analysis based on the response of the tested parents and F_2_ populations to stripe and leaf rusts at the adult plant stage, under field conditions. Data illustrated in Figure 9 and Figure 10 indicate that the two wheat genotypes, Sids-14 and Shandweel-1, consistently expressed susceptibility to stripe and leaf rusts. While, the three wheat parents showed varied levels of resistance to stripe and leaf rusts. For the twelve crosses, the frequency distribution of disease severity of F_2_ plants ranged from 0 to 80%. The resistant F_2_ plants were selected and the transferred genes were detected. The three genes, *Lr34/Yr18*, *Lr37/Yr17,* and *Lr67/Yr46*, were identified in F_2_ plants as fragments 220, 285, and 198 bp, respectively (Figure 11). Therefore, these results can be used to reduce the resistance breeding period for these genotypes. Finally, the grain yield of selected plants was evaluated.

## 4. Discussion

Breeding programs are one of the most cost-effective strategies to control stripe and leaf rusts. Wheat breeding programs all over the world are incorporating rust resistance genes into commercial cultivars [3]. A successful wheat breeding program depends mainly on the types and genetic variability available in wheat genotypes. The genetic variability is the most important natural resource in providing the required traits to develop new cultivars [4]. Genetic analysis has been extensively used to determine the gene action and system controlling the quantitatively inherited characters [17,18]. Qualitative resistance mediated by a single resistant gene, also known as major gene resistance (MGR), and race specific resistance are the two main types of host-genetic resistance. Quantitative resistance is controlled by a large number of minor genes for resistance, which have an additive impact. Adult plant resistance (APR), race non-specific, slow-rusting resistance, and partial resistance (PR) are all synonyms [19].

Responses of 23 wheat genotypes were evaluated against the most aggressive and frequent races of stripe rust (6E4, 159E255, and 250E254) and leaf rust (STSJT, MTTGT, and TTTST) at seedling and adult stages. At seedling stage, six wheat genotypes (Misr-3, Misr-4, Giza-171, Sakha-95, Gemmeiza-12, and Sids-14) were resistant to all the tested races and showed low infection type for the two diseases. While, two wheat genotypes (Misr-1 and Misr-2) were susceptible against all the tested races and showed high infection type of stripe rust. These results differed from Abdelaal et al. [20], who showed that Misr-1 and Misr-2 were resistant at seedling stage during the 2012/13 and 2013/14 seasons. This result may be attributed to the appearance of new virulent races such as 159E255 and 250E254, which were capable of supplanting the resistance of these genotypes. The most aggressive races in breaking the resistance in wheat genotypes (Misr-1, Misr-2, Gemmeiza-5, Gemmeiza-7, Gemmeiza-9, and Sids-12) were 250E254 and 159E255. Moreover, race TTTST of leaf rust was the most aggressive race. It was able to break the resistance in 14 wheat genotypes. Since rust inoculum comes in Egypt each year from outside sources and is moved from one region to another in the same year, this work should be continued [21].

Under field conditions, the four wheat genotypes Misr-3, Misr-4, Gemmeiza-12, and Gia-171 displayed strong and high levels of adult plant resistance of the two diseases under study. Misr-1, Misr-2, Gemmeiza-7, and Sids-12 showed the highest FRS (%) and AUDPC values of stripe rust. This result was inconsistent with Abdelaal et al. [20] who showed that Misr-1 and Misr-2 were resistant at adult stage during the 2012/13 and 2013/14 seasons. Moreover, Esmail et al. [22] reported that Misr-1 and Misr-2 were resistant at Nubaria and Kafrelsheikha locations during the 2017/18 season. The reason for supplanting the resistance could be attributed to the appearance of new races of the pathogen [23]. Additionally, Gemmeiza-9 and Gemmeiza-10 exhibited the highest FRS and AUDPC values of leaf rust. In contrast, Abdelbacki et al. [24,25] mentioned that Gemmeiza-9 and Gemmeiza-10 were resistant at adult stage, during the 2010/2011 and 2011/2012 seasons. This is due to the dynamic nature of the pathogen that led to continuous emergence of new aggressive races, such as 250E254 and TTTST of stripe and leaf rusts, respectively, which were able to overcome the newly deployed resistance genes in the released wheat genotypes: Sakha-95, Sids-14, and Shandweel-1 [26,27].

It was necessary to explain the resistance of the genotypes under study as a result of the resistance genes which play an important role in the durability of stripe and leaf rusts resistance in the most cultivated wheat. The findings of the current study specifically show that molecular markers outperform pedigree data in determining the role of resistance genes in wheat genotypes, which is consistent with numerous research and reviews [28]. Slow rusting or partial resistance has been shown to last longer than single seedling resistance [29]. The use of molecular markers established for most genes of stripe and leaf rusts resistance simplifies the pyramiding of these genes. Future host selection pressure on the pathogen might be reduced further by rotating genes across time and geography or by deploying cultivars with different effective resistance genes in various regions. However, traditional genetic and molecular marker studies will be required to confirm and extend the current results about the *Yr* and *Lr* genes, which are important for stripe and leaf rusts resistance in wheat genotypes in both seedlings and adults. As a result, using resistant genotypes to manage this serious disease is the most effective and environmentally sustainable approach. Therefore, incorporating pathogen resistance genes into adapted genotypes is a key aim in most wheat resistance breeding programs.

The marker for *Lr34/Yr18* was identified as a fragment of 220 bp in 10 genotypes (Misr-3, Misr-4, Misr-2, Misr-1, Sids-13, Gemmeiza-12, Sakha-94, Giza-168, Gia-171, and Giza-139), whereas it was not detected in nine genotypes (Sids-12, Sids-14, Sakha-61, Sakha-95, Gemmeiza-5, Gemmeiza-7, Gemmeiza-9, Gemmeiza-10, and Shandweel-1). This gene was characterized as a slow rusting resistance gene and reported to induce durable resistance than the single seedling resistance [29]. The leaf rust resistance genes (*Lr13* and *Lr34*) were identified in three wheat genotypes (Tijereta, E. Halco, and E. Calandria), according to German and Kolmer [30]. Likewise, *Lr46/Yr29* was considered as slow rusting, which explains why certain genotypes are vulnerable to infection while still producing a high yield [31]. *Lr46/Yr29* was detected in all the tested genotypes except Gemmeiza-7. The leaf rust resistance gene *Lr*37 is closely linked with *Yr*17 and *Sr38* [32]. It was identified as a fragment of 285 bp in four genotypes (Misr-4, Misr-3, Gemmeiza-12, and Giza-171). The marker for *Lr67/Yr46* was detected in 16 genotypes. Herrera-Foessel et al. [33] had previously discovered slow rusting resistance genes for leaf and stripe rusts in wheat, *Lr67,* and *Yr46*.

It can be concluded that genotypes Misr-3, Misr-4, Giza-171, and Gemmeiza-12 carry the four genes under study. They fall into one group through the polygenic tree. This result explains the reason for resistance in these genotypes for stripe and leaf rusts and recommended the use of these genotypes as a parental in stripe and leaf rusts resistance breeding programs. These findings further highlight the significance of pyramiding genes as a method for achieving long-term tolerance in low genetic diversity, high gene flow, and asexual mating systems [34]. Therefore, combining multiple successful resistance genes into a single genotype could help increase the resistance duration.

The resistance of newly released wheat genotypes such as Sids-14 and Shandweel-1 has rapidly lost its potency and these genotypes become susceptible in a short period due to the sudden emergency of the new and more aggressive races of *P. striiformis* f. sp. *tritici* and *P. triticina* f. sp. *tritici* [35]. Therefore, Sids-14 and Shandweel-1 wheat genotypes with varying degrees of stripe and leaf rusts severity and six F_2_ plants obtained from half diallel crosses between these genotypes and slow rusting genes *Lr34/Yr18*, *Lr37/Yr17*, and *Lr67/Yr46* for each disease were used in this research. The resistant F_2_ plants were selected and the transferred genes were detected. The three genes: *Lr34/Yr18*, *Lr37/Yr17,* and *Lr67/Yr46* were identified in F_2_ plants as fragments 220, 285, and 198 bp. Thus, these results can be used to reduce the resistance breeding period for these genotypes.

All of the tested genotypes had similarity between their pedigrees. Therefore, the identity of the rust resistance genes is essential for the incorporation of new effective resistance genes into wheat breeding programs and maintenance diversity of resistance genes in new released wheat genotypes. As a result, plant breeders should not only depend on the host pedigree, but also consider the pathogen genotype and environment as two key factors for disease development. Genetic resistance is the most cost-efficient and successful strategy for minimizing yearly production losses and preventing catastrophic epidemics. Due to the obvious dynamic nature of rust pathogens, which allows it to develop new virulent races that may disintegrate or surpass the host genetic resistance, plant breeders must continually introduce new effective resistance genes to their breeding materials. Therefore, having more knowledge about the genetic nature and inheritance of rust resistance is critical for establishing an important first step towards fully using and exploiting this resistance in wheat breeding programs and making the right decisions.

## 5. Conclusions

Under field conditions, the four wheat genotypes, Misr-3, Misr-4, Gemmeiza-12, and Gia-171 displayed strong and high levels of adult plant resistance. This result was confirmed by detecting more than one gene of resistance to stripe and leaf rusts to achieve high levels of resistance in these genotypes. Moreover, the resistance was improved in two new wheat genotypes, Sids-14 and Shandweel-1, by introgression of more than one gene. This information will be used to make an adequate decision in the future and to plan ahead for a rust-resistant wheat breeding program.

## Figures and Tables

**Figure 1 jof-07-00622-f001:**
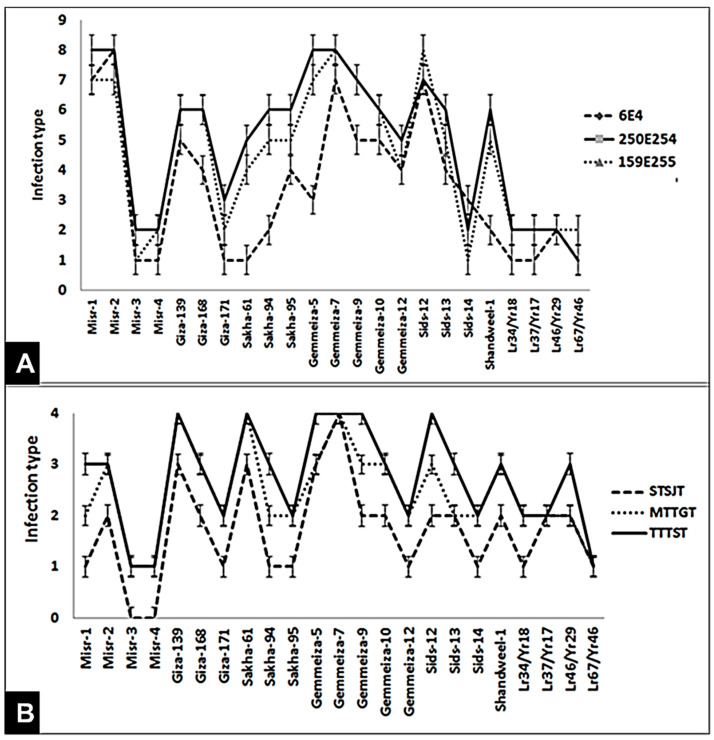
Seedling response of 23 wheat genotypes against the three races of stripe rust (**A**) and leaf rust (**B**) at seedling stage under greenhouse conditions.

**Figure 2 jof-07-00622-f002:**
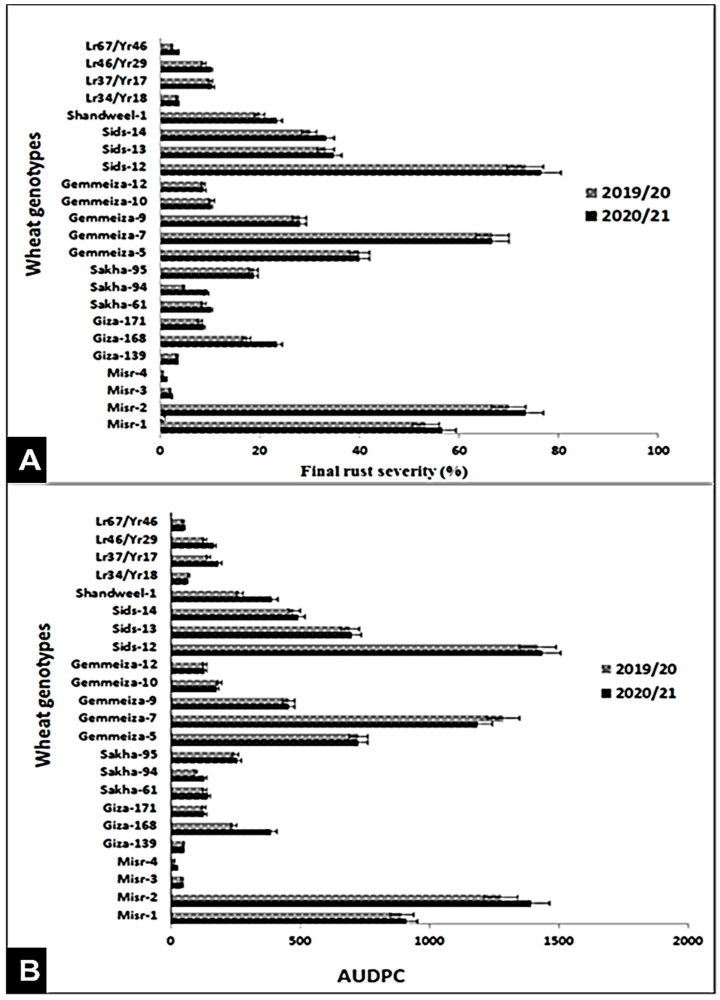
Final rust severity (%) (**A**) and AUDPC (**B**) of 23 wheat genotypes against stripe rust under field conditions at Kafr El-Sheikh governorate, during the 2019/20 and 2020/21 growing seasons.

**Figure 3 jof-07-00622-f003:**
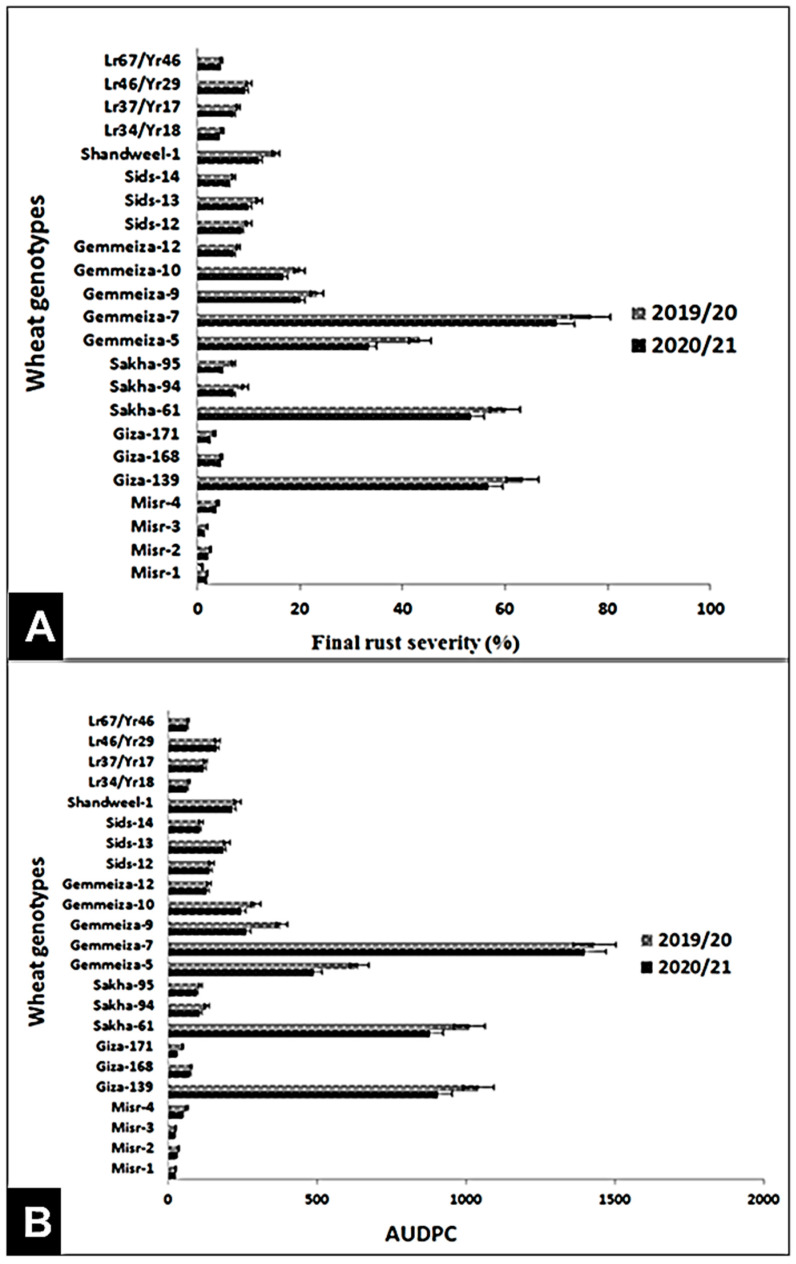
Final rust severity (%) (**A**) and AUDPC (**B**) of 23 wheat genotypes against leaf rust under field conditions at Kafr El-Sheikh governorate, during the 2019/20 and 2020/21 growing seasons.

**Figure 4 jof-07-00622-f004:**
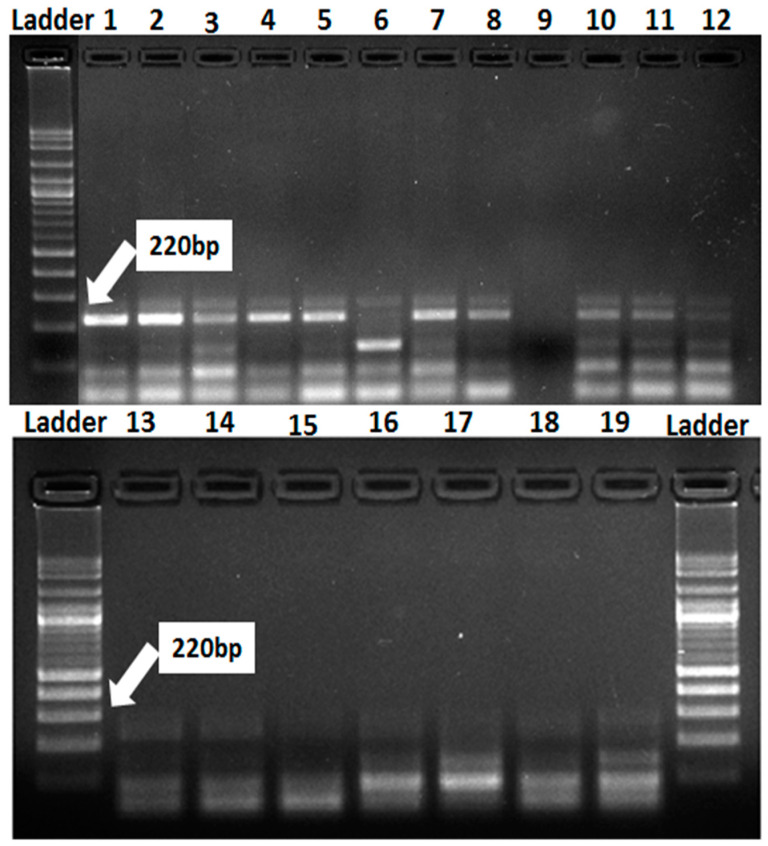
Electrophoretic amplified pattern of DNA extracted from 19 wheat genotypes using the specific primer for *Lr34/Yr18*. Lane 1 = Misr-3, Lane 2 = Misr-4, Lane 3 = Misr-2, Lane 4 = Misr-1, Lane 5 = Sids-13, Lane 6 = Gemmeiza-9, Lane 7 = Gemmeiza-12, Lane 8 = Sakha-94, Lane 9 = Sakha-95, Lane 10 = Giza-168, Lane 11 = Gia-171, Lane 12 = Giza-139, Lane 13 = Sids-12, Lane 14 = Sakha-61, Lane 15 = Sids-14, Lane 16 = Gemmeiza-5, Lane 17 = Gemmeiza-7, Lane 18 = Gemmeiza-10, and Lane 19 = Shandweel-1.

**Figure 5 jof-07-00622-f005:**
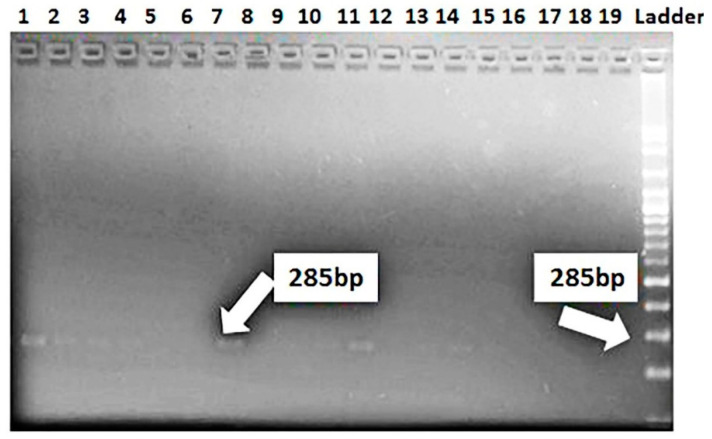
Electrophoretic amplified pattern of DNA extracted from 19 wheat genotypes using the specific primer of *Lr37/Yr17*. Lane 1 = Misr-3, Lane 2 = Misr-4, Lane 3 = Misr-2, Lane 4 = Misr-1, Lane 5 = Sids-13, Lane 6 = Gemmeiza-9, Lane 7 = Gemmeiza-12, Lane 8 = Sakha-94, Lane 9 = Sakha-95, Lane 10 = Giza-168, Lane 11 = Gia-171, Lane 12 = Giza-139, Lane 13 = Sids-12, Lane 14 = Sakha-61, Lane 15 = Sids-14, Lane 16 = Gemmeiza-5, Lane 17 = Gemmeiza-7, Lane 18 = Gemmeiza-10, and Lane 19 = Shandweel-1.

**Figure 6 jof-07-00622-f006:**
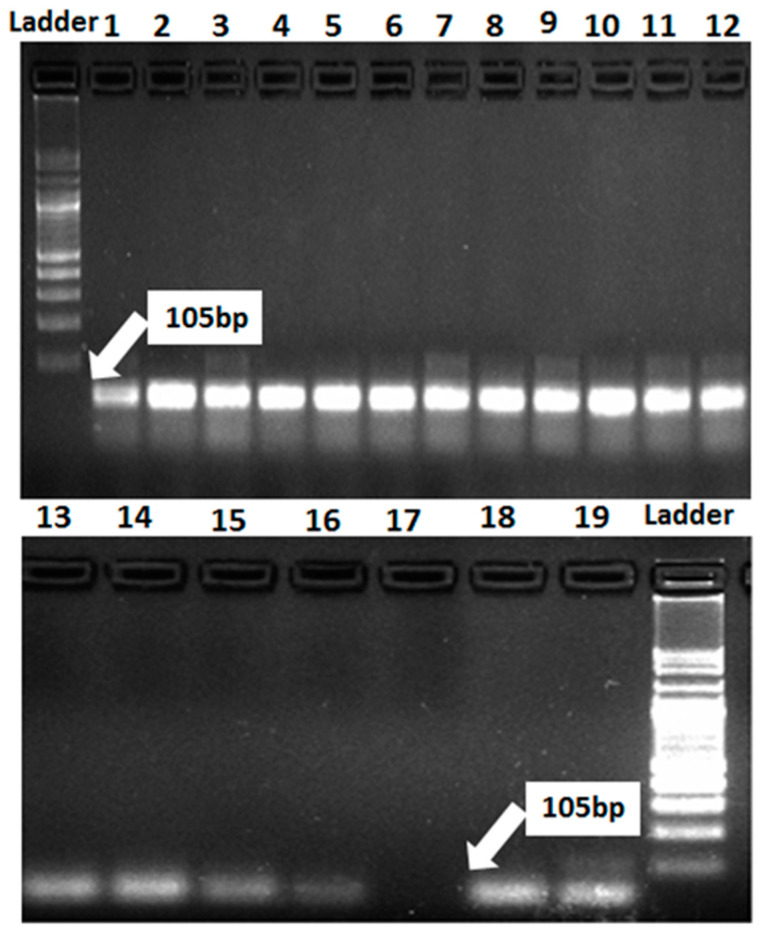
Electrophoretic amplified pattern of DNA extracted from 19 wheat genotypes using the specific primer of *Lr46/Yr29*. Lane 1 = Misr-3, Lane 2 = Misr-4, Lane 3 = Misr-2, Lane 4 = Misr-1, Lane 5 = Sids-13, Lane 6 = Gemmeiza-9, Lane 7 = Gemmeiza-12, Lane 8 = Sakha-94, Lane 9 = Sakha-95, Lane 10 = Giza-168, Lane 11 = Gia-171, Lane 12 = Giza-139, Lane 13 = Sids-12, Lane 14 = Sakha-61, Lane 15 = Sids-14, Lane 16 = Gemmeiza-5, Lane 17 = Gemmeiza-7, Lane 18 = Gemmeiza-10, and Lane 19 = Shandweel-1.

**Figure 7 jof-07-00622-f007:**
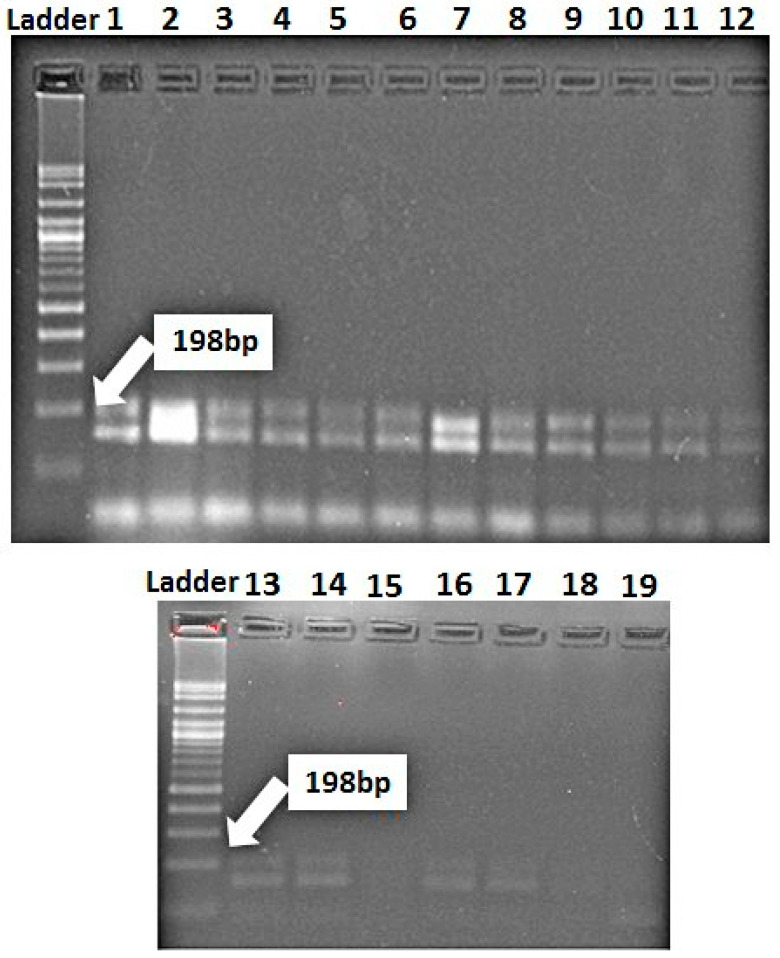
Electrophoretic amplified pattern of DNA extracted from 19 wheat genotypes using the specific primer of *Lr67/Yr46*. Lane 1 = Misr-3, Lane 2 = Misr-4, Lane 3 = Misr-2, Lane 4 = Misr-1, Lane 5 = Sids-13, Lane 6 = Gemmeiza-9, Lane 7 = Gemmeiza-12, Lane 8 = Sakha-94, Lane 9 = Sakha-95, Lane 10 = Giza-168, Lane 11 = Gia-171, Lane 12 = Giza-139, Lane 13 = Sids-12, Lane 14 = Sakha-61, Lane 15 = Sids-14, Lane 16 = Gemmeiza-5, Lane 17 = Gemmeiza-7, Lane 18 = Gemmeiza-10, and Lane 19 = Shandweel-1.

**Figure 8 jof-07-00622-f008:**
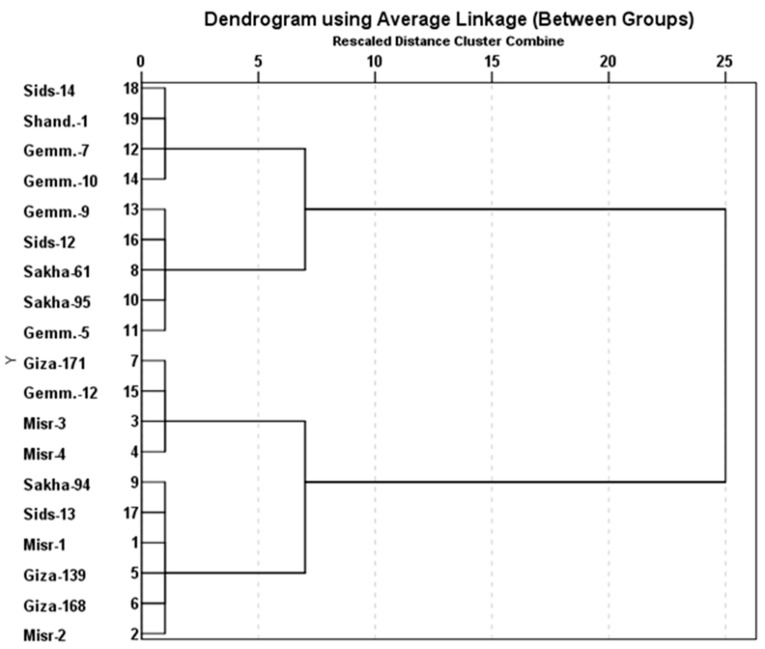
The polygenic tree of 19 wheat genotypes using the identified genes.

**Figure 9 jof-07-00622-f009:**
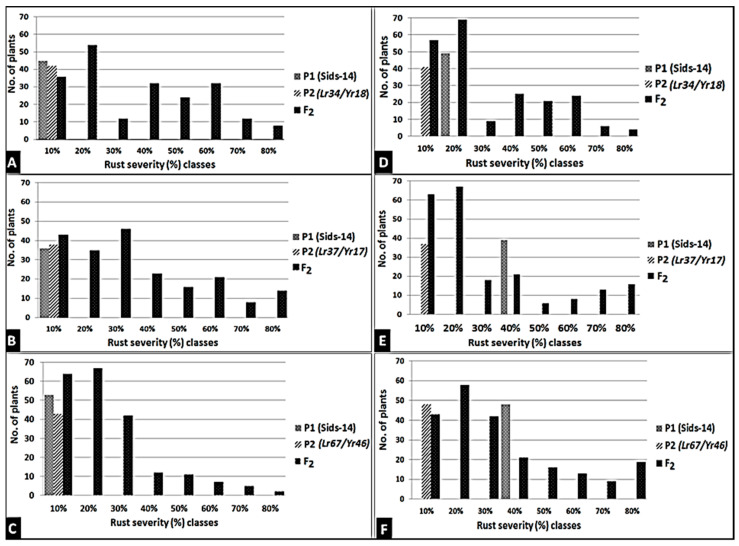
Frequency distribution of stripe and leaf rust severities (%) to Sids-14 × *Lr34/Yr18* (**A**), Sids-14 × *Lr37/Yr17* (**B**), and Sids-14 × *Lr67/Yr46* (**C**) for leaf rust and Sids-14 × *Lr34/Yr18* (**D**), Sids-14 × *Lr37/Yr17* (**E**), and Sids-14 × *Lr67/Yr46* (**F**) for stripe rust.

**Figure 10 jof-07-00622-f010:**
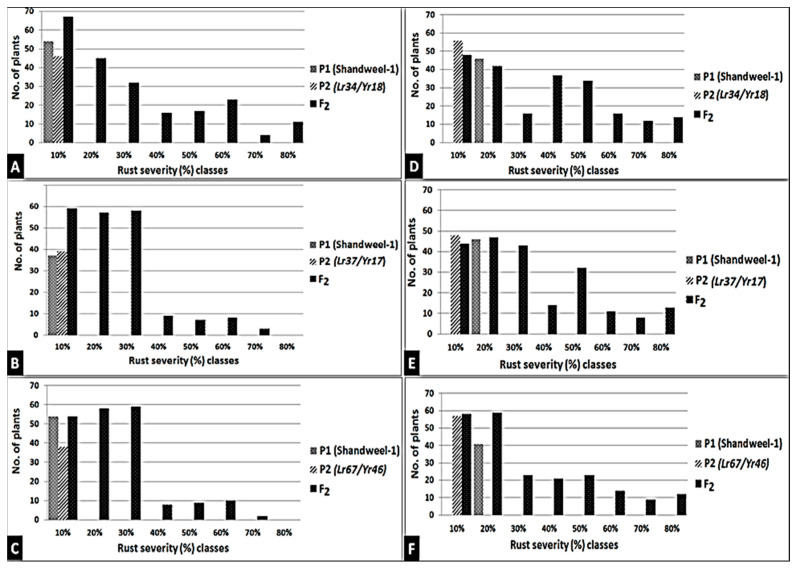
Frequency distribution of stripe and leaf rust severities (%) to Shandweel-1 × *Lr34/Yr18* (**A**), Shandweel-1 × *Lr37/Yr17* (**B**), and Shandweel-1 × *Lr67/Yr46* (**C**) for leaf rust and Shandweel-1 × *Lr34/Yr18* (**D**), Shandweel-1 × *Lr37/Yr17* (**E**), and Shandweel-1 × *Lr67/Yr46* (**F**) for stripe rust.

**Figure 11 jof-07-00622-f011:**
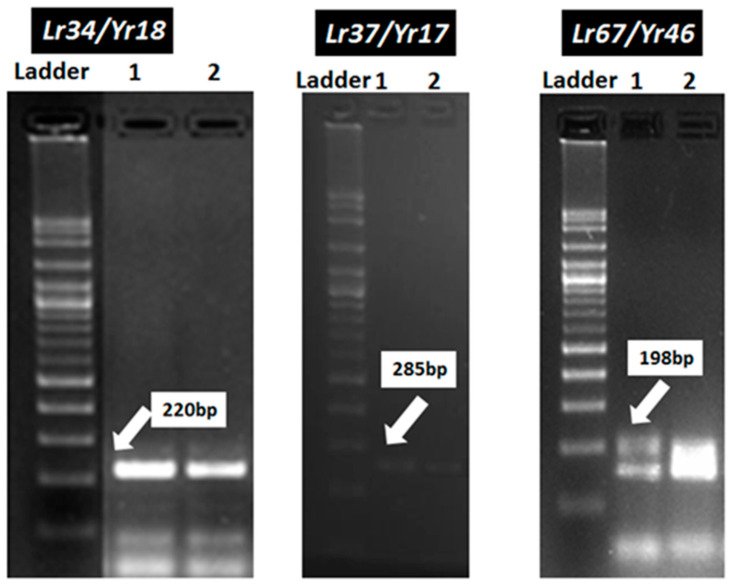
Electrophoretic amplified pattern of DNA extracted from six F_2_ crosses using the specific primer of *Lr34/Yr18*, *Lr37/Yr17* and *Lr67/Yr46*. Lane 1 = cross Sids-14, Lane 2 = cross Shandweel-1.

**Table 1 jof-07-00622-t001:** List of the tested wheat genotypes that were used in this study.

No.	Genotype	Pedigree
1	Misr-1	OASIS/SKAUZ//4*BCN/3/2*PASTOR.CMSSOYO1881T-050M-030Y-O3OM-30WGY-33M-0Y-0S.
2	Misr-2	SKAUZ/BAV92. CMSS96M0361S-1M-010SY-010M-010SY-8M -0Y-0S.
3	Misr-3	ATTILA*2/PBW65*2//KACHU CMSS06Y00582T-099TOPM-099Y-099ZTM-099Y-099M-10WGY-0B-0EGY
4	Misr-4	NS-732/HER/3/PRL/SARA//TSI/VEE#5/FRET2/5/WHEAR/SOKOLL
5	Giza-139	HINDI90/KENYA256G.
6	Giza-168	MIL/BUC//Seri CM93046-8M-0Y-0M-2Y-0B
7	Giza-171	Sakha 93/Gemmeiza 9 S.6-1GZ-4GZ-1GZ-2GZ-0S
8	Sakha-61	INIA/RL4220//7CYR”S”. CM15430-2S-2S-0S-0S.
9	Sakha-94	OPATA/RAYON//KAUZ. CMBW90Y3280-0TOPM-3Y-010M-010M-010Y-10M-015Y-0Y-0AP-0S.
10	Sakha-95	PASTOR//SITE/MO/3/CHEN/AEGILOPS SQUARROSA(TAUS)//BCN/4/WBLL1CMSA01Y00158S-040P0Y-040M-030ZTM-040SY-26M-0Y-0SY-0S
11	Gemmeiza-5	VEE”S”/SWM6525. GM4017-1GM-6GM-3GM-0GM.
12	Gemmeiza-7	CMH74A.630/SX//SER182/3/AGENT. GM4611-2GM-3GM-1GM -0GM.
13	Gemmeiza-9	ALD”S”/HUAC”S”//CMH74A.630/SX. GM4583-5GM-1GM-0GM.
14	Gemmeiza-10	MAYA74”S”/0N//160-147/3/BB/GLL/4/CHAT”S”/5/CROW”S”. GM5820-3GM-1GM-2GM-0GM.
15	Gemmeiza-12	OTUS/3/SARA/THB//VEE. CCMSS97Y00227S-5Y-010M-010Y -010M-2Y-1M-0Y-0GM
16	Sids-12	BUC//7C/ALD/5/MAYA74/ON//1160-147/3/BB/GLL/4/CHAT”S”/6/MAYA/VUL-4SD-1SD-1SD-0SD.
17	Sids-13	KAUZ “S”//TSI/SNB”S”. ICW94-0375-4AP-2AP-030AP-0APS-3AP-0APS-050AP-0AP-0SD.
18	Sids-14	SW8488*2/KUKUNACGSS01Y00081T-099M-099Y-099M-099B-9Y-0B-0SD.
19	Shandaweel-1	SITE/MO/4/NAC/TH.AC//3*PVN/3/MIRLO/BUC CMSS93B00567S-72Y-010M-010Y-010M-3Y-0M-0THY-0SH
20	Lr34/Yr18	TC*6/P158548(RL6058)
21	Lr37/Yr17	TC*6/VPM (RL6081)
22	Lr46/Yr29	Pavon 76
23	Lr67/Yr46	RL6077

**Table 2 jof-07-00622-t002:** Names, sequences, and references of specific primers linked to the tested genes used in this study.

Gene	Marker	Sequence of Primers 5′–3′	Fragment Size	Reference
*Lr34/Yr18*	Cslv34 F	GTT GGT TAA GAC TGG TGA TGG	220	Lagudah et al. 2006
	Cslv34R	GTG TTG CGC AAG TTT GTG A		
*Lr37/Yr17*	Ventriup	AGGGGCTACTGACCAAGGCT	285	Helguera et al. 2003
	LN2	TGCAGCTACAGCAGTATGTACACAAAA		
*Lr46/Yr29*	Xgwm259F	AGG GAA AAG ACA TCT TTT TTT TC	105	William et al. 2003
	Xgwm259R	CGA CCG ACT TCG GGT TC		
*Lr67/Yr46*	CFD71F	CAA TAA GTA GGC CGG GAC AA	198	Forrest et al. 2014
	CFD71R	TGT GCC AGT TGA GTT TGC TC		

## Data Availability

Not applicable.

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
