# Peer review of "Wheat Resistance to Stripe and Leaf Rusts Conferred by Introgression of Slow Rusting Resistance Genes"

_jof, 2021, doi:10.3390/jof7080622_

Round 1
Reviewer 1 Report
Please, carefully examine the corrections and the suggestions provided in the pdf file to improve the scientific quality of the manuscript.
Attention needs to be paid for the following:
- Many terms in plant pathology were misused (e.g., virulence, tolerance, disease incidence vs. disease severity, races of the disease, and many others highlighted in the pdf file).
- The Results section should mention only the data obtained, avoiding any discussion which should be transferred to the Discussion section.
- The Discussion needs to be improved. The paragraph looks like a literature review.
- The figures need to have their design improved for clarity. The use of letters in each graphic will help the reader follow the information easily.
- English needs to be reviewed to make the reading more pleasurable. Several grammatical errors were found.

Author Response
Dear Sir,
Thank you very much for the valuable revision to our manuscript submitted to the Journal of fungi. Attached here is the revised version of the manuscript amended according to the suggestions of the referees. All changes were highlighted. Please also find our point-by-point response to their comments.
Thank you very much.
Sincerely,
Prof. Mohsen Elsharkawy
Many terms in plant pathology were misused (e.g., virulence, tolerance, disease incidence vs. disease severity, races of the disease, and many others highlighted in the pdf file).
Done. All the suggested changes were done according to reviewers comments.
The Results section should mention only the data obtained, avoiding any discussion which should be transferred to the Discussion section.
Done.
The Discussion needs to be improved. The paragraph looks like a literature review.
Done. We summarized the first paragraph and kept only the necessary information that is needed in explaining the results.
The figures need to have their design improved for clarity. The use of letters in each graphic will help the reader follow the information easily.
Done. All figures were amended according to your valuable suggestions. Thank you
English needs to be reviewed to make the reading more pleasurable. Several grammatical errors were found.
Done. The manuscript was reviewed, and all unnecessary sentences were deleted, and the whole manuscript was improved.
Thank you very much for your valuable revision.

Reviewer 2 Report
Manuscript ID: jof-1306013
Title: “Wheat resistance to stripe and leaf rusts conferred by introgression of slow rusting resistance genes”
General
This is an interesting study and the authors have collected a unique dataset concerning the subject. The paper is generally well written and structured. Below I have provided numerous remarks to improve the text language. In several instances, I also suggested to cite more relevant and recent literature.
Introduction
Line 35: stable CORRECT TO staple
Line 37: due to CORRECT TO because of
Lines 49 and 50: A limited genetic base, in comparison to broad genetic diversity, is a serious im-49 pediment to breeding for various biotic and abiotic stresses REPLACE BY Compared to broad genetic diversity, a limited genetic base is a serious im-49 pediment to breeding for various biotic and abiotic stresses.
Lines 51 – 54: However, wheat cultivars developed with larger genetic bases are very effective in improving the yield under several agro-climatic environments and resist the spread of diseases in new released cultivars [5]. REPLACE BY However, wheat cultivars developed with larger genetic bases effectively improve the yield under several agro-climatic environments and resist t the main objectives he spread of diseases in new released cultivars [5].
Line 55: more than CORRECT TO over
Line 62: due to CORRECT TO because of
Line 68: Furthermore, genomic DELETE Furthermore
Line 69: possible use DELETE possible
Line 75: In general, using DELETE In general
Line 76: Consequently, incorporating DELETE Consequently
Line 78: the main objectives COPRRECT TO the major objectives
Materials and Methods
Line 83: were conducted COPRRECT TO was conducted
Line 85: at adult COPRRECT TO at an adult
Line 86, 87: Additionally, the molecular DELETE Additionally
Line 95: seven-days-old COPRRECT TO seven-day-old
line 98: strip rust CORRECT TO stripe rust
Line 99: While, they were envaulted CORRECT TO While they were evaluated
Line 104: On the other hand, data of leaf rust DELETE On the other hand
Line 111: belt CORRECT TO belts
Line 120: Area CORRECT TO The area
Line 135: 30s CORRECT TO the 30s
Line 147: production CORRECT TO the production
Line 152: the two CORRECT TO two
Results
Line 210: high infection type CORRECT TO acute infection type
Line 210: On the other hand, the rest of wheat CORRECT TO The rest of the wheat
Line 212: was able to CORRECT TO could
Line 220: adult stage CORRECT TO the adult stage
Line 225: On the other hand, Misr-1, CORRECT TO Misr-1,
Line 232: On the other hand, Giza-139, CORRECT TO Giza-139,
Line 237: On the other hand, Sakha-94 CORRECT TO Sakha-94
Line 238: adult stage CORRECT TO the adult stage
Line 253: important role CORRECT TO an important role
Line 254: clearly indicate CORRECT TO clearly show
Line 255: evaluating the presence of resistance genes CORRECT TO evaluating resistance genes
Line 269: On the other hand, DELETE
Lines 279, 180: On the other hand, DELETE
Discussion
Line 370: Breeding programs CORRECT TO breeding programs
Line 402: On the other hand DELETE
Line 416: play important role CORRECT TO which play an important role
Figures
Figure 5: It is not clear at 285 bp

Author Response
Dear Sir,
Thank you very much for the valuable revision to our manuscript submitted to the Journal of Fungi. Attached here is the revised version of the manuscript amended according to the suggestions of the referees. Please also find our point-by-point response to their comments.
Thank you very much.
Sincerely,
Prof. Mohsen Elsharkawy
Introduction
Line 35: stable CORRECT TO staple
DONE.
Line 37: due to CORRECT TO because of
DONE.
Lines 49 and 50: A limited genetic base, in comparison to broad genetic diversity, is a serious im-49 pediment to breeding for various biotic and abiotic stresses REPLACE BY Compared to broad genetic diversity, a limited genetic base is a serious im-49 pediment to breeding for various biotic and abiotic stresses.
DONE.
Lines 51 – 54: However, wheat cultivars developed with larger genetic bases are very effective in improving the yield under several agro-climatic environments and resist the spread of diseases in new released cultivars [5]. REPLACE BY However, wheat cultivars developed with larger genetic bases effectively improve the yield under several agro-climatic environments and resist t the main objectives he spread of diseases in new released cultivars [5].
DONE.
Line 55: more than CORRECT TO over
DONE.
Line 62: due to CORRECT TO because of
DONE.
Line 68: Furthermore, genomic DELETE Furthermore
DONE.
Line 69: possible use DELETE possible
DONE.
Line 75: In general, using DELETE In general
DONE.
Line 76: Consequently, incorporating DELETE Consequently
DONE.
Line 78: the main objectives COPRRECT TO the major objectives
DONE.
Materials and Methods
Line 83: were conducted COPRRECT TO was conducted
DONE.
Line 85: at adult COPRRECT TO at an adult
DONE.
Line 86, 87: Additionally, the molecular DELETE Additionally
DONE.
Line 95: seven-days-old COPRRECT TO seven-day-old
DONE.
line 98: strip rust CORRECT TO stripe rust
DONE.
Line 99: While, they were envaulted CORRECT TO While they were evaluated
DONE.
Line 104: On the other hand, data of leaf rust DELETE On the other hand
DONE.
Line 111: belt CORRECT TO belts
DONE.
Line 120: Area CORRECT TO The area
DONE.
Line 135: 30s CORRECT TO the 30s
DONE.
Line 147: production CORRECT TO the production
DONE.
Line 152: the two CORRECT TO two
DONE.
Results
Line 210: high infection type CORRECT TO acute infection type
DONE.
Line 210: On the other hand, the rest of wheat CORRECT TO The rest of the wheat
DONE.
Line 212: was able to CORRECT TO could
DONE.
Line 220: adult stage CORRECT TO the adult stage
DONE.
Line 225: On the other hand, Misr-1, CORRECT TO Misr-1,
DONE.
Line 232: On the other hand, Giza-139, CORRECT TO Giza-139,
DONE.
Line 237: On the other hand, Sakha-94 CORRECT TO Sakha-94
DONE.
Line 238: adult stage CORRECT TO the adult stage
DONE.
Line 253: important role CORRECT TO an important role
DONE.
Line 254: clearly indicate CORRECT TO clearly show
DONE.
Line 255: evaluating the presence of resistance genes CORRECT TO evaluating resistance genes
DONE.
Line 269: On the other hand, DELETE
DONE.
Lines 279, 180: On the other hand, DELETE
DONE.
Discussion
Line 370: Breeding programs CORRECT TO breeding programs
DONE.
Line 402: On the other hand DELETE
DONE.
Line 416: play important role CORRECT TO which play an important role
DONE.
Figures
Figure 5: It is not clear at 285 bp
We improved the figure to make it clear.
